# Mucosal Addressin Cell Adhesion Molecule-1 Mediates T Cell Migration into Pancreas-Draining Lymph Nodes for Initiation of the Autoimmune Response in Type 1 Diabetes

**DOI:** 10.3390/ijms252111350

**Published:** 2024-10-22

**Authors:** Yankui Li, Rachel C. Gunderson, Zeyu Xu, Wenjia Ai, Fanru Shen, Jiayu Ye, Baohui Xu, Sara A. Michie

**Affiliations:** 1Department of Vascular Surgery, Tianjin Medical University Second Hospital, Tianjin 300211, China; 2Department of Pathology, School of Medicine, Stanford University, Stanford, CA 94305, USA; rcook@gmail.com (R.C.G.); xu2zy@ucmail.uc.edu (Z.X.); wenjiaai@stanford.edu (W.A.); baohuixu@stanford.edu (B.X.); smichie@stanford.edu (S.A.M.); 3Department of Surgery, School of Medicine, Stanford University, Stanford, CA 94305, USA; 4Department of Medicine, College of Medicine, University of Cincinnati, Cincinnati, OH 45219, USA; 5Department of Medicine, School of Medicine, Stanford University, Stanford, CA 94305, USA; jyye@stanford.edu

**Keywords:** type 1 diabetes, T cell migration, adhesion molecule, regulation T cells

## Abstract

Type 1 diabetes (T1D) is an autoimmune disease that is caused by autoreactive T cell-mediated destruction of insulin-producing β cells in the pancreatic islets. Although naive autoreactive T cells are initially primed by islet antigens in pancreas-draining lymph nodes (pan-LNs), the adhesion molecules that recruit T cells into pan-LNs are unknown. We show that high endothelial venules in pan-LNs of young nonobese diabetic mice have a unique adhesion molecule profile that includes strong expression of mucosal addressin cell adhesion molecule-1 (MAdCAM-1). Anti-MAdCAM-1 antibody blocked more than 80% of the migration of naive autoreactive CD4^+^ T cells from blood vessels into pan-LNs. Transient blockade of MAdCAM-1 in young nonobese diabetic mice led to increased numbers of autoreactive regulatory CD4^+^ T cells in pan-LNs and pancreas and to long-lasting protection from T1D. These results indicate the importance of MAdCAM-1 in the development of T1D and suggest MAdCAM-1 as a potential therapeutic target for treating T1D.

## 1. Introduction

Migration of T cells from the bloodstream into organs is essential for the development of protective immunity to foreign pathogens. In the initiation stage of the immune response, naive T cells, which have migrated through blood vessel high endothelial venules (HEVs) into secondary lymphoid organs, including lymph nodes (LNs) and Peyer’s patches (PPs), are primed by antigen-bearing dendritic cells [1]. In the effector stage, the progeny of the primed T cells migrate from blood vessels into the target organ, such as the skin or lung, where the pathogens exist. In each stage, migration must be tightly regulated so that appropriate subsets of lymphocytes detect and destroy foreign pathogens without causing excessive inflammation and tissue damage. This regulation is provided, in part, by multistep adhesion cascades in which vascular endothelial adhesion molecules in secondary lymphoid and target organs bind to their ligands on circulating lymphocytes [2,3]. The combinations of adhesion molecules that are expressed by endothelia differ from one organ to another and from uninflamed versus inflamed organs, thus targeting specific subsets of lymphocytes to specific sites [4,5,6,7,8,9,10,11,12,13].

The selective targeting of T cells to secondary lymphoid and target organs is also essential for the development of organ-specific T cell-mediated autoimmune diseases, such as type 1 diabetes (T1D) [14]. Nonobese diabetic (NOD) mice are a model for human autoimmune type 1 diabetes, characterized by the progressive destruction of insulin-secreting β cells by the infiltration of autoreactive T cells in pancreatic islets with early onset and a high rate of diabetes in female mice (https://www.jax.org/strain/001976, accessed on 15 October 2024). Studies on NOD mice indicate that the autoimmune response is initiated when naive autoreactive T cells migrate through HEVs into pancreas-draining lymph nodes (pan-LNs), where they are primed by islet antigen-bearing dendritic cells at 3 weeks of age [15,16,17,18,19]. In the effector stage, memory/effector T cells migrate from the bloodstream into the pancreas, leading to progressive inflammation of the islets, destruction of the β cells and eventually the development of T1D [20,21]. We and others have shown that mucosal addressin cell adhesion molecule-1 (MAdCAM-1) is highly expressed on vascular endothelia in inflamed islets of NOD mice and is important for recruiting α_4_β_7_ integrin^+^ lymphocytes into the islets and for the development of T1D [4,8,22,23,24]. In contrast, deficiency of L-selectin, which is the major lymphocyte receptor for endothelial peripheral node addressin (PNAd), does not affect the development of T1D in NOD mice [25,26]. However, the adhesion molecules that control the migration of naive autoreactive T cells from blood vessels into pan-LNs in the initiation stage of T1D are unknown.

The purpose of this study was to determine which HEV adhesion molecules are critical for the migration of naive T cells into pan-LNs and, thus, for the initiation of the T cell autoimmune response in NOD mice. First, we determined which adhesion molecules are expressed by HEVs in pan-LNs of young NOD mice. Next, we used in vivo lymphocyte migration assays to define the physiologic roles of highly expressed HEV adhesion molecules and their lymphocyte ligands in the migration of naive T cells into pan-LNs. Last, we asked if transient blockade of the most robust HEV adhesion molecule in young NOD mice affects the priming of naive autoreactive T cells and the development of islet inflammation and T1D.

## 2. Results

### 2.1. MAdCAM-1 and PNAd Are Expressed on HEVs in Pan-LNs

The critical window for initial priming of naive autoreactive T cells in pan-LNs of NOD mice is 3 to 4 weeks of age [15]. Since almost all naive lymphocytes enter LNs through HEVs [27], we determined which adhesion molecules were expressed on pan-LN HEVs of 3–4-week-old female NOD mice. We focused on HEV adhesion molecules that mediate organ-selective migration of lymphocytes into secondary lymphoid organs. These include MAdCAM-1, which binds to lymphocyte α_4_β_7_ integrin and L-selectin [9,28], PNAd, which binds to lymphocyte L-selectin [10,29,30], and vascular cell adhesion molecule 1 (VCAM-1), which binds to lymphocyte α_4_β_1_ integrin [7].

We found co-expression of MAdCAM-1 and PNAd on almost every HEV in pan-LNs (Figure 1a,b), intestine-draining LNs (int-LNs) (Figure 1d,e), and small intestine PPs (Figure 1g,h). In contrast, most HEVs in skin-LNs expressed PNAd but little or no MAdCAM-1 (Figure 1j,k). Negative control mAbs (Figure 1c,f,i,l) and an anti-VCAM-1 mAb did not stain HEV endothelia in any of the organs.

In adult mice, PNAd shows strong diffuse expression on skin-draining LN (skin-LN) HEVs and is important for adhesion and recruitment of circulating L-selectin^+^ naive lymphocytes. In contrast, PNAd is found mainly on the abluminal side of PP HEVs, where it is unavailable to bind to and thus unable to mediate the migration of intravascular L-selectin^+^ lymphocytes into PPs [10,31,32]. Given that MAdCAM-1 was strongly expressed on almost every HEVs of pan-LNs, int-LNs and PPs, and absent or rare on HEVs of skin-LNs, we thus determined the pattern of PNAd expression on HEVs in pan-LNs of young NOD mice compared to skin-LNs, int-LNs, and PPs. The three patterns, strong diffuse (black arrow), abluminal (black arrowhead), and mixed (white arrow), are illustrated in Figure 1b,e,h,k. In int-LNs and skin-LNs, more than 80% of the PNAd^+^ HEVs had strong diffuse expression (Table 1). In contrast, 85% of the PNAd^+^ HEVs in PPs showed abluminal expression. No single pattern predominated in pan-LNs: 17% of the PNAd^+^ HEVs were strongly diffuse, 26% were mixed, and 57% were abluminal (Table 1). Thus, HEVs in pan-LNs of young NOD mice have an adhesion molecule profile (MAdCAM-1^+^, PNAd^+^ with variable expression pattern), which is significantly different from that of HEVs in int-LNs (MAdCAM-1^+^ with almost strong diffuse PNAd expression), PPs (MAdCAM-1^+^ with almost weak abluminal PNAd expression), and skin-LNs (strong diffuse PNAd expression with no or rare MAdCAM-1^+^).

### 2.2. Naive T Cells Migrate Efficiently from Blood Vessels into Pan-LNs

To evaluate the ability of individual subsets of T cells to migrate from the bloodstream into pan-LNs, lymphocytes from the spleen and LNs of 3–4-week-old NOD mice were labeled with Tetramethyl-rhodamine-5(6)-isothiocyanate (TRITC) and transferred intravenously (iv) into age-matched NOD mice. More than 95% of the donor T cells were naive (CD44^low^ CD45RB^high^). Two hours after donor cell transfer, there were relatively more CD4^+^ donor T cells in host pan-LNs than in skin-LNs, int-LNs and PPs (Figure 2a). In contrast, CD8^+^ T cells migrated equally well to all LNs and poorly to PPs (Figure 2b). NOD/BDC2.5 naive autoreactive CD4^+^ T cells, which express a transgenic Vβ4 T cell receptor that recognizes an islet antigen [33], migrated into pan-LNs significantly better than into skin-LNs, int-LNs, and PPs (Figure 2c). Thus, NOD naive CD4^+^ T cells (most of which are not specific for islet antigens) and NOD/BDC2.5 naive autoreactive CD4+ T cells migrate most efficiently into pan-LNs of young NOD mice.

### 2.3. MAdCAM-1 Is a Dominant Endothelial Adhesion Molecule for Migration of Naive T Cells into Pan-LNs

To identify the adhesion molecules that direct the migration of naive T cells into pan-LNs as compared to int-LNs, skin-LNs and PPs, we performed short-term in vivo lymphocyte migration assays. Anti-MAdCAM-1 mAb blocked more than 80% of the migration of naive T cells from young NOD donors and naive autoreactive CD4^+^ T cells from NOD/BDC2.5 donors into pan-LNs, int-LNs and PPs of young NOD hosts, without affecting the homing of young NOD donor T cells and naive autoreactive CD4^+^ T donor cells into skin-LNs (Figure 3a,b). In contrast, anti-PNAd mAb almost completely blocked the homing of young NOD donor T cells and naive autoreactive CD4^+^ T donor cells into skin-LNs but impaired less than 40% of the migration of either T cell population into pan-LNs (Figure 3c,d).

Anti-α_4_β_7_ integrin mAb treatment (Figure 3f) blocked approximately 60%, 75% and >80% of the migration of NOD T cells into pan-LNs, int-LNs, and PPs, respectively, without affecting T cell homing to skin-LNs (Figure 3e). In contrast, anti-L-selectin mAb treatment blocked 70%, >95%, 75% and approximately 50% of the migration of NOD donor T cells into pan-LNs, skin-LNs, int-LNs and PPs, respectively (Figure 3f). Similarly, the migration of T cells from 4–5 weeks old NOD β_7_ integrin^−/−^ mice into wild-type (WT) host pan-LNs, int-LNs and PPs were reduced by approximately 60%, 60%, and 90%, respectively, without impacting T cell migration to skin-LNs (Figure 3g). T cells from 4–5 weeks old NOD β_7_ integrin^−/−^ L-selectin^−/−^ mice entered pan-LNs, int-LNs, skin-LNs, and PPs more than 90% less efficiently than did cells from NOD WT mice (Figure 3h). Thus, similar to int-LNs and PPs but distinct from skin-LNs, MAdCAM-1 is the dominant functional vascular addressin that mediates the migration of naive T cells from blood vessels into pan-LNs of young NOD mice.

### 2.4. Blockade of MAdCAM-1 in 3-Week-Old NOD Mice Prevents Development of Diabetes but Not Islet Inflammation

To test the hypothesis that temporarily blocking MAdCAM-1 during the critical window for priming of naive autoreactive T cells would inhibit the development of islet inflammation and diabetes, we gave 3-week-old NOD mice one intraperitoneal (ip) injection of anti-MAdCAM-1 or isotype control mAb. At 5 and 10 weeks of age, there was little difference in the degree of islet inflammation between treatment groups (Figure 4a). While mean inflammation scores of 7-week-old negative control mAb-treated mice were higher than those of 7-week-old anti-MAdCAM-1-treated mice, the difference was not statistically significant. Although blockade of MAdCAM-1 failed to prevent islet inflammation, none of 10 mice treated with anti-MAdCAM-1 mAb developed diabetes within 1 year of treatment (Figure 4b). In contrast, 8 of 9 mice treated with control mAb became diabetic. Thus, temporary inhibition of MAdCAM-1 during the initiation stage of the autoimmune response provides significant, long-lasting protection from diabetes but not from islet inflammation.

### 2.5. Blockade of MAdCAM-1 in 3 Weeks Old NOD Mice Does Not Abolish Priming of Naive Autoreactive CD4^+^ T Cells in Pan-LNs

To examine the effects of MAdCAM-1 blockade on pan-LNs, we gave 3-week-old NOD mice one ip injection of anti-MAdCAM-1 mAb or control mAb. At 4 and 5 weeks of age, the absolute numbers of CD4^+^ T cells (Figure 5a) and CD8^+^ T cells (Figure 5b) in pan-LNs and int-LNs, but not skin-LNs or spleen, of anti-MAdCAM-1-treated NOD mice were significantly lower than those in control mAb-treated mice. At 8 weeks of age, there were no differences between treatment groups in the number of T cells in LNs (Figure 5a,b) or spleen.

To evaluate the effects of MAdCAM-1 blockade on autoreactive T cells, we transferred CFSE-labeled lymphocytes from young NOD/BDC2.5 mice into anti-MAdCAM-1- or control mAb-treated 3-week-old NOD mice. More than 95% of the donor autoreactive T cells were naive. One week after anti-MAdCAM-1 or control mAb treatment, the absolute number of donor-derived autoreactive CD4^+^ T cells in pan-LNs of anti-MAdCAM-1 mAb-treated mice was significantly less than in control mAb-treated mice (Figure 5c). However, there was no difference between treatment groups in the percentage of donor-derived autoreactive CD4^+^ T cells in the total lymphocytes in pan-LNs. Moreover, the proportion of donor-derived autoreactive CD4^+^ T cells in pan-LNs that had undergone proliferation (Figure 5d,e) or activation (CD44^high^ or CD69^+^, Figure 5f,g) did not differ significantly between groups. Thus, the MAdCAM-1 blockade did not prevent the priming of the autoreactive T cells that were able to enter pan-LNs.

### 2.6. Blockade of MAdCAM-1 in Young NOD and NOD/BDC2.5 Mice Causes a Relative Increase in Islet Antigen-Specific Tregs in Pan-LNs and Inflamed Pancreas

Our results thus far indicate that transient blockade of MAdCAM-1 prevents the development of diabetes but not activation of naive autoreactive T cells in pan-LNs or development of islet inflammation. Treg cells are critical for the regulation of autoimmune responses in T1D [17,34,35,36,37,38]. To determine if MAdCAM-1 blockade alters autoantigen-specific Treg cellularity in pan-LNs, we transferred lymphocytes from young NOD/BDC2.5 donor mice into anti-MAdCAM-1- or control mAb-treated 3-week-old NOD host mice. One week thereafter, anti-MAdCAM-1 mAb treatment had no significant impact on the relative numbers of host CD4^+^ T cells (Figure 6a, left) or donor CD4^+^ T cells (Figure 6a, right) in pan-LNs of host mice. However, forkhead box protein P3 (Foxp3), which is a Treg marker, was expressed on 19% of the donor-derived proliferating autoreactive CD4^+^ T cells in pan-LNs of anti-MAdCAM-1-treated mice, as compared to 11% in control mAb-treated mice (Figure 6b; *p* < 0.05). There were no differences between groups, however, in the expression of Foxp3 on nonproliferating donor autoreactive CD4^+^ T cells (Figure 6b) or host CD4^+^ T cells.

Next, we used flow cytometry to evaluate CD4^+^ lymphocytes from the pancreata of 16-week-old NOD mice (there were not enough pancreatic lymphocytes in younger mice for analysis) and 5-week-old NOD/BDC2.5 mice that had been treated with mAb at 3 weeks of age. There were relatively fewer CD4^+^ T cells in the pancreata of anti-MAdCAM-1-treated NOD and NOD/BDC2.5 mice than in the pancreata of control mAb-treated mice (Figure 6c). However, the MAdCAM-1 blockade led to the enrichment of the pancreatic CD4^+^Foxp3^+^ CD25^+^ cells (Figure 6d,e). Thus, blockade of MAdCAM-1 is associated with a relative increase in Tregs in both pan-LNs and pancreas.

## 3. Discussion

Organ-draining LNs are the sites where tissue-specific immune response is initiated in response to foreign and autoantigens. In the NOD mouse model of T1D disease, several previous studies, including LN resection at different ages, have demonstrated that pan-LNs are critical for priming naive autoreactive T cells by autoantigen-bearing antigen-presenting cells with the age of 3 weeks as the critical time window for initiating autoimmunity to pancreatic islet antigens [15,16,17,18,19]. Therefore, the migration of naive autoreactive T cells to pan-LNs is crucial for triggering T1D autoimmunity in NOD mice. Although the recognition of vascular addressins by cognate lymphocyte receptors (adhesion molecules) guides the migration of lymphocytes to secondary lymphoid tissues such as LNs, it has not been defined which HEV adhesion molecule(s) are expressed on HEVs of, and important for the recruitment of naive autoreactive T cells into, pan-LNs for the initiation of T1D the autoimmune response to islet antigens. In the present study, we found that pan-LN HEVs in 3–4-week-old NOD mice express both MAdCAM-1 and PNAd. However, the expression pattern is quite distinct from that of skin-LN HEVs, which express high levels of PNAd with little or no MAdCAM-1. Though MAdCAM-1 and PNAd are co-expressed by most HEVs in int-LNs and PPs, these organs differ significantly from pan-LNs in the location of PNAd on the HEVs and in the importance of PNAd to lymphocyte recruitment. Specifically, we found strong diffuse expression of PNAd on most skin-LN HEVs and abluminal expression of PNAd on most PP HEVs, as reported for adult mice [10,31,32]. In contrast, neither pattern was predominant on pan-LN HEVs. Thus, pan-LN HEVs have a unique adhesion molecule phenotype.

Because only luminal, but not abluminal, expression of a vascular addressin is available to bind to cognate receptors on, and thus mediates the homing of, circulating lymphocytes into tissues, the expression patterns of MAdCAM-1 and PNAd on HEVs determine the functional significance of each addressin in the migration of circulating lymphocytes into LNs and PPs in NOD mice. As expected from the expression patterns of MAdCAM-1 and PNAd in different LNs and PPS, our short-term in vivo lymphocyte migration studies revealed that MAdCAM-1 played a predominant role in the migration of naive T cells from blood vessels into pan-LNs, int-LNs and PPs, but not skin-LNs, in 3–4 weeks old NOD mice. In contrast, strong diffuse PNAd on skin-LN HEVs accounted for the migration of almost naive donor T cells to skin-LNs in NOD host mice, while PNAd only mediated the migration of approximately 40% of donor naive T cells into pan-LNs due to >50% of pan-LN HEVs with abluminal PNAd expression. Additionally, predominant abluminal expression of PNAd on PP HEV had no role in the migration of naive donor T cells to host PPs. We and others have shown that MAdCAM-1 is highly expressed on HEVs in inflamed islets of NOD mice and is important for the migration of T cells into the islets and B cells to pan-LNs [4,8,12,22,23,24]. Thus, MAdCAM-1 unifies the migration of lymphocytes into pan-LNs and inflamed pancreatic islets in NOD mice.

Although we previously reported that MAdCAM-1 and its lymphocyte ligand α_4_β_7_ integrin mediate the migration of B cells to pan-LNs of young NOD mice [12], there are no studies that directly examine the role of MAdCAM-1 in the migration of T cells into pan-LNs in the NOD mouse model of T1D [39]. However, indirect evidence suggests the potential involvement of MAdCAM-1 and α_4_β_7_ integrin in the migration of T cells into pan-LNs. For example, Hanninen et al. found that a T cell line established from the pancreas of a T1D patient adhered avidly to vessels in pan-LNs, inflamed islets and appendix but poorly to vessels in skin-LNs [40]. Adoptively transferred islet antigen-specific T cells from a clone established from a T1D patient have been shown to accumulate in pancreas and pan-LNs but not skin-LNs in NOD-severe combined immunodeficiency mice [41]. Moreover, in some patients with T1D, peripheral blood T cells that react with the putative islet autoantigen glutamic acid decarboxylase 65 express high levels of α_4_β_7_ integrin, a major ligand for MAdCAM-1 [42]. Thus, additional studies are required to evaluate the potential role(s) of MAdCAM-1 in the development of human T1D.

Here, we show that a single injection of anti-MAdCAM-1 mAb at 3 weeks of age prevented T1D in NOD mice with equal or greater efficacy than did long-term blockade of MAdCAM-1 as reported previously [24,43]. While one injection of anti-MAdCAM-1 mAb resulted in transient hypocellularity in pan-LNs and int-LNs, it failed to abrogate the proliferation and activation of autoreactive T cells that entered pan-LNs. Although cells from int-LNs of prediabetic NOD mice also adoptively transferred diabetes [44], we and others have shown that initial activation of autoreactive T cells occurs in pan-LNs rather than int-LNs [15,16,45].

Despite completely preventing T1D, transiently blockade of MAdCAM-1 in our studies did not prevent the development of islet inflammation, suggesting the initiation of immunosuppressive mechanisms. Accumulating evidence indicates that Treg cells potently suppress autoimmunity in NOD mice. For example, NOD strains with few or no Treg cells have accelerated/severe autoimmune disease [36,37]. Moreover, T1D in NOD mice can be suppressed or reversed by adoptive transfer of in vitro expanded autoreactive Treg cells or by therapeutic interventions, such as administration of anti-CD3 mAb and/or oral or nasal islet antigen, that increase the number and suppressive function of Tregs [34,35,46,47,48]. We found that blockade of MAdCAM-1 augmented the induction of autoreactive Foxp3^+^ Treg cells in pan-LNs. Although anti-MAdCAM-1 mAb treatment did not impact the activation and proliferation of naive autoreactive CD4^+^ T cells in pan-LNs, we cannot completely exclude that these Treg cells exert a subtle effect on autoreactive T cells in pan-LNs, as reported previously [17]. In support of our findings, reconstitution of Treg deficient NOD/BDC 2.5-Foxp3^sf^ mice with Treg cells failed to abolish the activation or proliferation of T cells in pan-LNs [37].

In our study, MAdCAM-1 blockade led to increased numbers of Treg cells in the pancreata of NOD and NOD/BDC 2.5 mice. Although islet inflammation develops in NOD/BDC 2.5 and NOD/BDC 2.5 Foxp3^sf^ strains similarly, it is extremely aggressive in the Foxp3^sf^ mice and quickly leads to diabetes, suggesting that the function of Treg cells is primarily confined to islets where they combat aggressive effector cells [37]. Thus, we assume that the Treg cells in islets after MAdCAM-1 blockade may exert direct and/or indirect effects on autoreactive effector/memory T cells in the pancreas and thus prevent the progression of insulitis and the development of diabetes. These Treg cells may have been recruited from the bloodstream and/or may have converted in situ from conventional memory CD4^+^ T cells [49].

A humanized anti-CD3 mAb (Tzield, teplizumab-mzwv) has been recently approved by the USA Food and Drug Administration (FDA) to treat individuals who are more than 8 years old and have stage 2 type 1 diabetes (https://www.accessdata.fda.gov/drugsatfda_docs/label/2022/761183s000lbl.pdf, accessed on 16 October 2024) [50,51]. Teplizumab treatment has been shown to induce Tregs in both human and humanized animals [52,53,54,55,56]. In addition to the impact of MAdCAM-1 blockage on the T cellularity and Tregs of pan-LN and inflamed islets in NOD mice, MAdCAM-1 was also expressed on endothelial cells and mediated the homing of β_7_ integrin-expressing hematopoietic stem cells into bone marrow [57]. Thus, the suppression of T1D by temporary blockage of MAdCAM-1 in NOD mice may be mediated by inhibiting the migration of lymphocytes into pan-LNs and inflamed islets, increasing Tregs in inflamed islets and potentially impairing hematopoietic stem cell migration to bone marrow. Thus, targeting MAdCAM-1 may be more effective than teplizumab.

Two recombinant humanized anti-MAdCAM-1 mAbs, PF-00547659 and SHP64/Ontamlimab, are used for treating patients with inflammatory bowel disease (ulcerative colitis and Crohn’s disease) [58,59,60,61,62]. Vedolizumab/ENTYVIO, which is a humanized mAb against α_4_β_7_ integrin, a cognate lymphocyte receptor for MAdCAM-1, has been approved by the US FDA for clinical management of patients with moderately to severely active ulcerative colitis (https://www.accessdata.fda.gov/drugsatfda_docs/label/2022/125476Orig1s046lbl.pdf, accessed on 16 October 2024). Given the importance of the MAdCAM-1/α_4_β_7_ integrin pathway in the migration of T cells to pan-LNs and inflamed pancreas in mouse models of T1D, it warrants testing the therapeutic efficacy of humanized MAdCAM-1 mAbs in a humanized model or a clinical trial for T1D.

In summary, we show that MAdCAM-1 is highly expressed in pan-LN HEVs in young NOD mice. Moreover, MAdCAM-1 mediated the migration of most naive T cells, including those that react with islet antigen, from blood vessels into pan-LNs. Transient blockade of MAdCAM-1 in 3-week-old NOD mice led to increased numbers of Treg cells in inflamed pancreata and to long-term protection from the development of diabetes. Thus, MAdCAM-1 may be a novel therapeutic target for limiting T1D progression.

## 4. Materials and Methods

### 4.1. Mice

NOD, NOD/BDC2.5, NOD β_7_ integrin^−/−^ (NOD.B6-*Itgb7^tm1Cgn^*/2LtJ) and NOD β_7_ integrin^−/−^ L-selectin^−/−^ (NOD.Cg-*Sell^tm1Tft^ Itgb7^tm1Cgn^*/LtJ) mice were bred and housed in Stanford Research Animal Facility. The original mouse breeders were NOD mice from Taconic Biosciences, Germantown, NY, USA and NOD β_7_ integrin^−/−^ and NOD β_7_ integrin^−/−^L-selectin^−/−^ mice from The Jackson Laboratory, Bar Harbor, ME, USA. NOD/BDC2.5 mice (from Dr. Hugh O. McDevitt, Stanford University, Stanford, CA, USA) are a transgenic NOD mouse strain that expresses the rearranged T cell receptor (TCR) α and β from the diabetogenic *H2-A^g7^* restricted autoreactive BDC2.5 CD4^+^ T cell clone (https://www.jax.org/strain/004460, accessed on 16 October 2024). Thus, almost all CD4^+^ T cells from 3–4 week-old NOD/BDC2.5 mice are naive autoreactive CD4^+^ T cells identified by TCR Vβ4 mAb and are used for tracking autoantigen-specific T cell immune responses. In NOD mice, T1D onset occurs early with a high incidence (90–100% by 30 weeks of age) in female compared to male (40–60% by 30–40 weeks of age) NOD mice (https://www.jax.org/strain/001976, accessed on 15 October 2024). In our NOD colony, peri-insulitis develops at 4–5 weeks of age, and ~90% of females are diabetic by 30 weeks. Thus, female NOD mice at 3–4 weeks of age were used for all experiments unless otherwise stated. Stanford University’s Administrative Panel of Laboratory Animal Care approved all studies.

### 4.2. Antibodies and Other Reagents

MAbs to PNAd (clone MECA-79) and L-selectin (MEL-14) (both from Dr. Eugene C. Butcher, Stanford University), α_4_β_7_ heterodimer (DATK-32), MAdCAM-1 (MECA-367) and VCAM-1 (MK2.7) (all from American Type Culture Collection Manassas, Manassas, VA, USA) were used for immunohistology and in vivo studies. Fluorochrome-biotinylated mAbs to CD3 (145-2C11), CD4 (GK1.5), CD8 (53-6.7), CD25 (PC61), CD45RB (C363-16A), CD44 (IM7), and CD69 (all from Biolegend, San Diego, CA, USA), Foxp3 (FJK-16s, eBiosciences, San Diego, CA, USA) and TCR Vβ4 (Caltag, Burlingame, CA, USA) were used for flow cytometric analysis. Additional reagents were biotin-anti-rat IgG antibody, peroxidase-anti-rat IgM antibody and peroxidase-streptavidin (all from Jackson ImmunoResearch Laboratories, West Grove, PA, USA), CFSE and TRITC (both from Molecular Probes, Eugene, OR, USA), and phycoerythrin (PE)-streptavidin (BD Biosciences, Milpitas, CA, USA).

### 4.3. Immunohistochemistry

Acetone-fixed frozen sections of pan-LNs, int-LNs (aka mesenteric LN), skin-LNs, including superficial inguinal, cervical, axillary, and brachial LNs) and PPs were sequentially incubated with anti-MAdCAM-1, anti-VCAM-1 or isotype-matched negative control mAb for 1 h at room temperature, phosphate-buffered saline (PBS), biotin-anti-rat IgG antibody in 2% normal mouse serum for 30 min, PBS, peroxidase-streptavidin for 30 min and 3,3′-diaminobenzidine-tetrahydrochloride (DAB)/hydrogen peroxide solution for 10 min. To stain PNAd, sections were sequentially incubated with anti-PNAd or isotype-matched negative control mAb, PBS, peroxidase-anti-rat IgM antibody in 2% normal mouse serum, and DAB/hydrogen peroxide solution. Slides were counterstained with methylene blue or hematoxylin, dehydrated, cover-slipped, and examined with a light microscope. For PNAd staining, HEVs were classified as strong diffuse positive, in which there was strong staining of every endothelial cell in the HEV; abluminal, in which staining was restricted to the basement membrane area that encircles the HEV; and mixed, in which some cells in the HEV were strong diffuse positive and others were abluminal. At least 39 pan-LN HEVs were evaluated on each mouse.

### 4.4. T Cell Migration into Pan-LNs

TRITC-labeled lymphocytes from LNs and spleens of 3–4-week-old NOD (>95% of T cells were CD44^low^ CD45RB^high^ naive phenotype) or 6-month-old NOD/BDC2.5 mice were transferred iv into 3–4 week old female NOD mice (5 × 10^7^ cells/mouse) as previously described [7,12,63,64]. Host mice were sacrificed 2 h after cell transfer. Lymphocyte suspensions were prepared from the LNs and PPs of host mice and stained with PE-Cy7-anti-CD3, fluorescein isothiocyanate (FITC)-anti-CD4, and APC-anti-CD8 mAbs to detect NOD donor CD4^+^ T cells and CD8^+^ T cells, or with PE-Cy7-anti-CD4, allophycocyanin (APC)-anti-CD44 and FITC-anti-CD45RB mAbs to detect NOD/BDC2.5 donor naive autoreactive T cells. Flow cytometry was used to determine the percentage of each donor subpopulation in at least 50,000 total cells in the lymphocyte scatter gate for each organ.

### 4.5. In Vivo Blocking of T Cell Migration by Anti-Adhesion Molecule mAbs

Short-term in vivo lymphocyte migration assays were performed as previously described [7,12,63,64]. Briefly, TRITC-labeled lymphocytes from 3–4-week-old NOD or 6-month-old NOD/BDC2.5 mice were transferred iv into 3–4-week-old female NOD mice (5 × 10^7^ cells/mouse). To block endothelial adhesion molecules, each host mouse was given 500 μg anti-MAdCAM-1, anti-PNAd or isotype-matched negative control mAb iv 30 min before transfer. Antibodies against MAdCAM-1 and PNAd at 500 µg have been shown to completely block the migration of adoptively transferred donor lymphocytes to PPs and skin-LNs, respectively, in adult NOD mice in short-term lymphocyte migration assay [7,12,63,64]. To block lymphocyte adhesion molecules, donor lymphocytes were incubated for 15 min with 10 μg/mL an anti–lymphocyte adhesion molecule or a control mAb before transfer. Host mice were sacrificed 2 h after cell transfer. Lymphocyte suspensions from LNs and PPs were stained with PE-Cy7-anti-CD3 mAb to detect NOD donor T cells or with PE-Cy7-anti-CD4, APC-anti-CD44, and FITC-anti-CD45RB mAbs to detect NOD/BDC2.5 donor naive autoreactive T cells. Flow cytometry was used to determine the percentage of each donor subpopulation in at least 50,000 total cells in the lymphocyte scatter gate for each organ. Results are presented as donor lymphocyte migration in the anti-adhesion molecule mAb treatment group as the percentage of that in the negative control mAb treatment group, which is designated as 100%.

### 4.6. In Vivo Migration of T Cells from β_7_ Integrin^−/−^ and β_7_ Integrin^−/−^ L-Selectin^−/−^ NOD Mice

Spleen and LN lymphocytes from 4–5-week-old WT, β_7_ integrin^−/−^, β_7_ integrin^−/−^L-selectin^−/−^ mice were labeled with TRITC or CFSE as previously described [7,12,63,64]. Fifty million TRITC-labeled β_7_ integrin^−/−^ or β_7_ integrin^−/−^L-selectin^−/−^ deficient cells and fifty million CFSE-labeled WT cells were transferred iv into each host mouse. Mice were sacrificed 2 h after cell transfer. Lymphocyte suspensions were stained with PE-Cy7-anti-CD3 mAb and analyzed by flow cytometry. For each organ, results are presented as migration of T cells from donor knockout mice as the percentage of that from donor WT mice, which is designated as 100%.

### 4.7. Analysis of Diabetes Incidence, Islet Inflammation and LN T Cell Cellularity

Female NOD or NOD/BDC2.5 mice at 3 weeks of age were injected ip with anti-MAdCAM-1 or control mAb (30 μg mAb/g body weight). Mice were screened for glycosuria twice a week using a glucose strip until 60 weeks of age, and diabetes was confirmed by blood glucose greater than 250 mg/dL [4,24].

To assess islet inflammation, mice were sacrificed at 5, 7, and 10 weeks of age. Pancreatic frozen sections (150 μm apart) were stained with hematoxylin and eosin. Islet inflammation was graded blindly as no inflammation (grade 0), mild peri-islet inflammation (<20 lymphocytes next to the islet; grade 1), severe peri-islet inflammation (≥20 lymphocytes next to the islet; grade 2), or insulitis (infiltration of lymphocytes into the islet; grade 3) [4,24]. At least 33 islets were evaluated for each pancreas. The mean islet inflammation score was calculated for each mouse.

To determine LN T cell cellularity, anti-MAdCAM-1 or control mAb-treated mice were sacrificed at 4, 5, and 8 weeks of age. Single-cell suspensions were prepared from LNs and spleen, counted using a hemacytometer, and stained with PE-Cy7-anti-CD3, PE-anti-CD4, and APC-anti-CD8 mAbs. For each organ, the absolute number of cells in each subset was determined by multiplying the fraction of cells expressing the subset-specific marker (from flow cytometry analysis) by the total number of cells as described [4,24].

### 4.8. Analysis of Autoreactive CD4^+^ T Cell Proliferation and Activation in Pan-LNs

Three-week-old female NOD mice were injected ip with anti-MAdCAM-1 or control mAb (30 μg mAb/g body weight). Thirty minutes thereafter, spleen and LN lymphocytes were prepared from 3–4-week-old NOD/BDC2.5 mice, labeled with 5 μM CFSE and transferred iv into mAb-treated 3-week-old NOD mice (2–3 × 10^7^ cells/mouse). One week later, mice were sacrificed, and cell suspensions from LNs and spleen were prepared: (1) to evaluate donor NOD/BDC autoreactive T cell proliferation based on CFSE intensity reduction; (2) to stain with biotin-anti-TCR Vβ4 (detected by PE-Cy7-streptavidin), Alexa Fluor 405-CD44, PE-CD69, and PE-Cy5.5-anti-CD4 or APC-Cy7-CD4 mAbs to evaluate the absolute cell number and activation of donor-derived autoreactive CD4^+^ T cells in host pan-LNs; (3) to stain donor-derived autoreactive Treg cells with biotin-anti-Vβ4 (detected by streptavidin-PE-Cy7), pacific blue-anti-CD4, APC-Cy7-anti-CD25 and APC-anti-Foxp3 mAbs according to the manufacturer’s instructions (eBioscience, San Diego, CA, USA). Data on at least 5000 donor-derived autoreactive CD4^+^ T cells per sample were acquired on an LSR II flow cytometer using FACSDiva^TM^ software (Version 9.0, BD Biosciences, Milpitas, CA, USA) and analyzed using FlowJo software (Version 10.10.0, Tree Star Inc, Ashland, OR, USA).

### 4.9. Analysis of Pancreatic Treg Cells

Three-week-old NOD and NOD/BDC2.5 mice were injected ip with anti-MAdCAM-1 or isotype control mAb as described above, sacrificed at 16 (NOD) or 5 (NOD/BDC2.5) weeks of age, and perfused with PBS through the left ventricle. Pancreata were removed, minced, and filtered through a 40 μM cell strainer. Lymphocytes were isolated using Lympholyte-Mammal per the manufacturer’s instruction (Cedarlane Laboratories, Burlington, ON, Canada) and stained for analysis of Treg cells as described above.

### 4.10. Statistical Analysis

All data on continuous variables were tested for data normality using the Shapiro–Wilk test and presented as mean ± standard deviation (SD). One-sample or two-sample unpaired Student’s *t*-test, or two-way analysis of variance (ANOVA) followed by two-sample comparison, were used to test the difference in continuous variables among groups. Given the potent inhibition efficacy of anti-MAdCAM-1 mAb, as shown in previous studies [4,24], 10 mice for each group were used to test whether one injection of anti-MAdCAM-1 mAb at 3 weeks of age prevents T1D onset. The log-rank test was employed to test the difference in T1D incidence rate between the two groups. *p* < 0.05 is considered statistically significant. All analyses were performed using the GraphPad Prism (Version 10.2.1, GraphPad Software, LLC, San Diego, CA, USA).

## Figures and Tables

**Figure 1 ijms-25-11350-f001:**
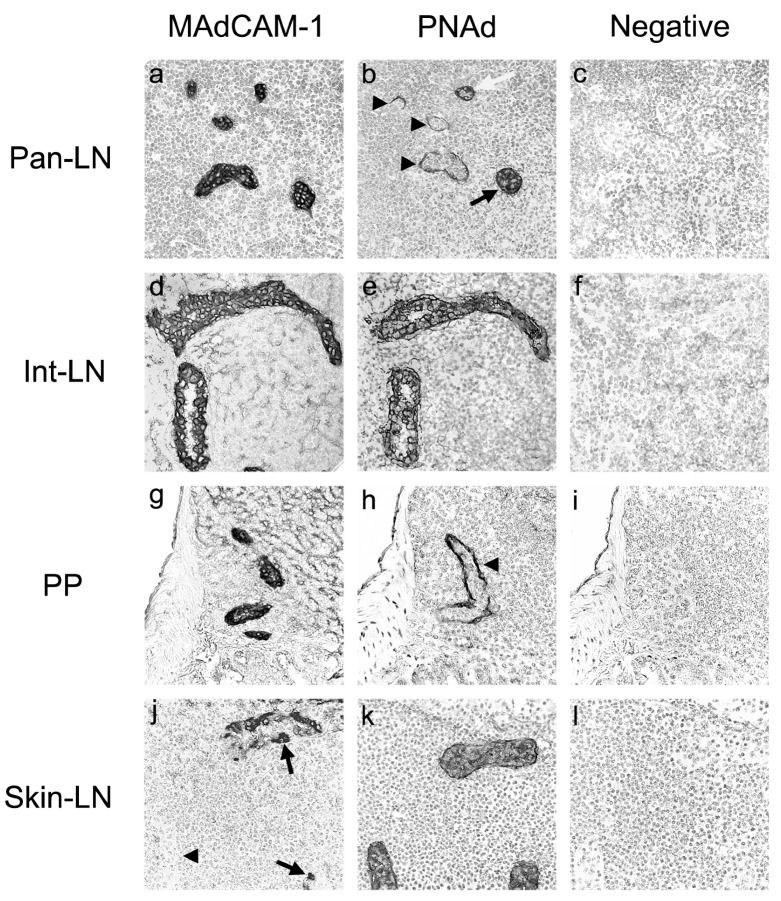
Representative staining of MAdCAM-1 and PNAd in HEVs of different tissue-draining lymph nodes (LNs) of young NOD mice. Frozen sections of pancreas-draining LNs (pan-LNs), intestine-draining LNs (int-LNs), small intestine PPs and skin-draining LNs (skin-LNs) from 3–4-week-old female NOD mice were stained with an anti-MAdCAM-1, anti-PNAd, or isotype-matched negative control monoclonal antibody (mAb). Strong MAdCAM-1 expression was noted on most HEVs in pan-LNs (**a**), int-LNs (**d**), and PPs (**g**) with focal (arrows) or no (arrowhead) MAdCAM-1 expression on skin-LN HEVs (**j**). Strong diffuse PNAd expression was seen on most HEVs in int-LNs (**e**) and skin-LNs (**k**), with abluminal expression of PNAd on most PP HEVs ((**h**), arrowhead). In pan-LNs (**b**). PNAd expression was noted in an abluminal pattern by many HEVs (arrowheads) and in a strong diffuse (black arrow) or mixed pattern (white arrow) by other HEVs. There was no staining of HEVs with isotype-matched negative control mAbs (**c**,**f**,**i**,**l**). Images in each row were the same tissue sections from the same mouse and stained with three different mAbs. Original magnification: 600×.

**Figure 2 ijms-25-11350-f002:**
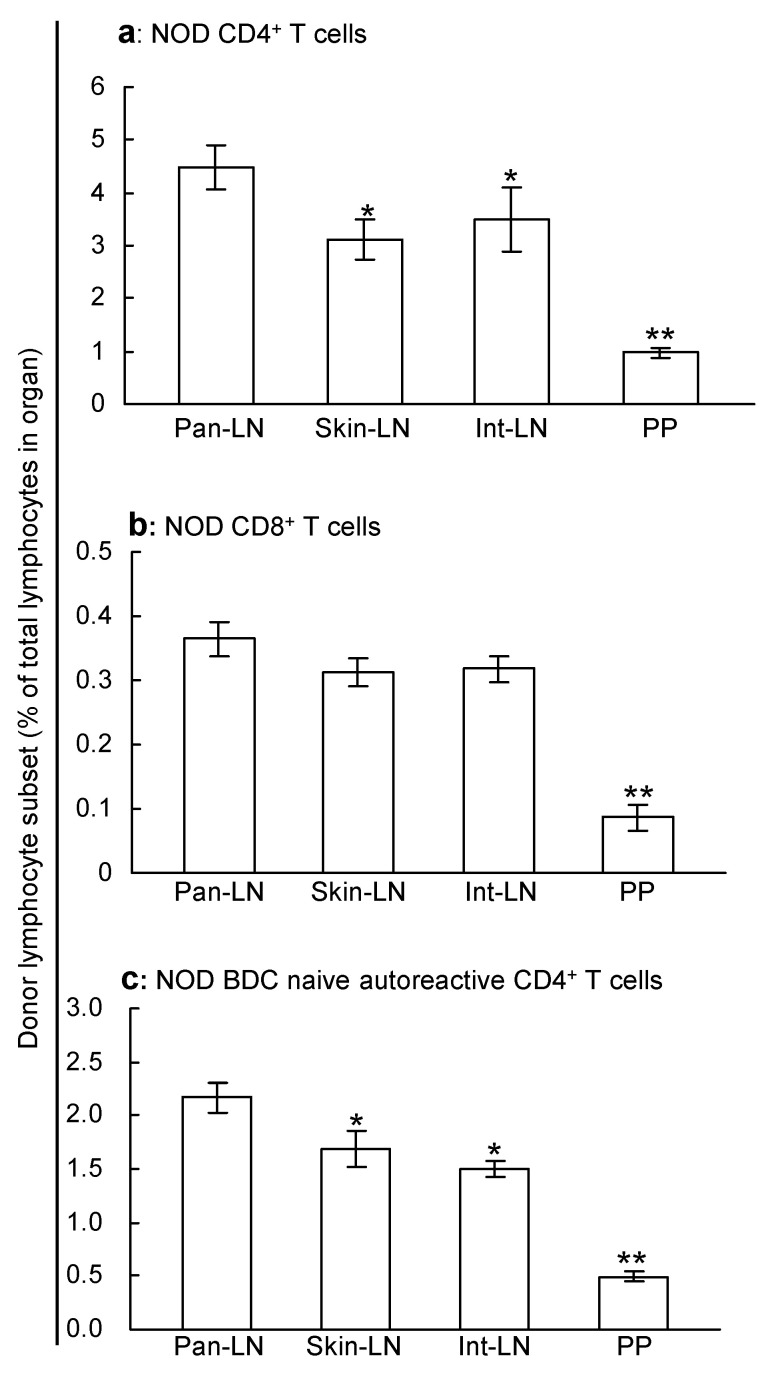
Naive T cells migrate efficiently from blood vessels into pancreas-draining lymph nodes (pan-LNs). Tetramethyl-rhodamine-5(6)-isothiocyanate (TRITC)-labeled lymphocytes from 3–4-week-old nonobese diabetic (NOD, (**a**,**b**)) or 6-month-old NOD/BDC2.5 mice (**c**) were transferred intravenoulsy into 3–4-week-old NOD mice. More than 95% of the NOD donor T cells were naive. Two hours thereafter, CD4^+^ T cells (**a**), CD8^+^ T cells (**b**), and naive autoreactive CD4^+^ T cells (**c**) were identified in host lymphoid organs by suspension immunofluorescent staining and flow cytometric analysis. Results are expressed as donor subset lymphocytes as the percentage of total lymphocytes in each organ. Data are mean ± standard deviation from 3–4 mice/group. Two-sample unpaired Student’s *t* test, * *p* < 0.05 and ** *p* < 0.01 compared to pan-LNs.

**Figure 3 ijms-25-11350-f003:**
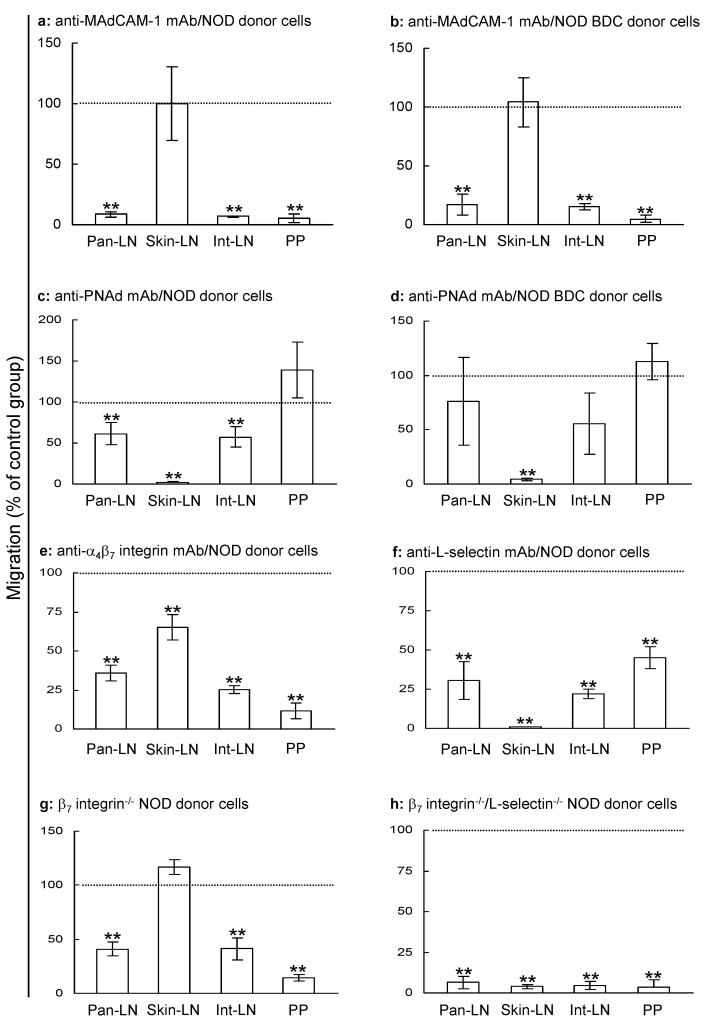
MAdCAM-1 plays a major role in the migration of naive T cells into pan-LNs of young nonobese diabetic (NOD) mice. In vivo lymphocyte migration assays were used to determine the physiologic significance of specific adhesion molecules in the migration of adoptively transferred lymphocytes from the bloodstream into pan-LNs of 3–4-week-old NOD host mice. (**a**–**d**): NOD wild type (WT) host mice were given a monoclonal antibody (mAb) against MAdCAM-1 (**a**,**b**), PNAd (**c**,**d**) or isotype-matched negative control mAb intravenously (iv, 500 µg/mouse) followed by iv transferring tetramethylrhodamine-5(6)-isothiocyanate (TRITC)-labeled lymphocytes from 3–4-week NOD (**a**,**c**) or 6-month-old NOD/BDC 2.5 mice (**b**,**d**). (**e**,**f**): TRITC-labeled lymphocytes from 3–4-week NOD mice were treated with mAb against α_4_β_7_ integrin (**e**), L-selectin (**f**) or isotype-matched negative control mAb (**e**,**f**) and transferred iv into NOD WT host mice. (**g**,**h**): NOD WT host mice were given a 1:1 mix of TRITC-labeled lymphocytes from NOD WT mice and 5(6)-carboxyfluorescein diacetate succinimide ester (CFSE)-labeled-lymphocytes from NOD β7 integrin^−/−^ mice (**g**) or NOD β_7_ integrin^−/−^L-selectin^−/−^ mice (**h**). All host mice were sacrificed 2 h after cell transfer. Donor T cells (**a**,**c**,**e**–**h**) or donor naive autoreactive CD4^+^ T cells (CD44^low^CD45RB^high^) (**b**,**d**) as a percentage of total lymphocytes in host LNs and PPs were determined using flow cytometry. Results are expressed as migration in the specific mAb treatment group as the percentage of that in the negative control mAb treatment group, in which the value is defined as 100% (horizontal dot-line) (**a**–**f**) or as migration of T cells from β_7_ integrin^−/−^ or β_7_ integrin^−/−^L-selectin^−/−^ donor mice as the percentage of that from WT donor mice (**g**,**h**). All data are mean ± standard deviation from 3–4 mice/group. One sample Student’s *t*-test, ** *p* < 0.01 compared to control mAb group (**a**–**e**,**f**) or WT lymphocytes (**g**,**h**), in which migration is set at 100%.

**Figure 4 ijms-25-11350-f004:**
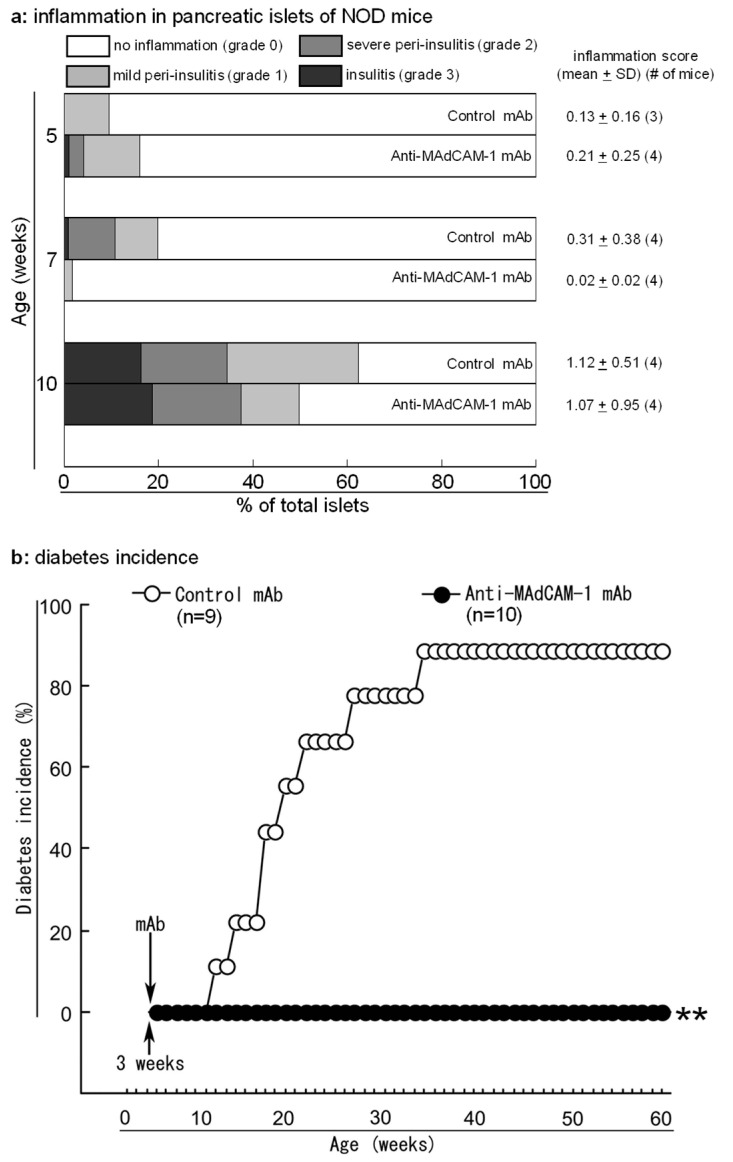
Blockade of MAdCAM-1 in young nonobese diabetic (NOD) mice blocks the development of diabetes but not islet inflammation. Three-week-old female NOD mice were given an intraperitoneal injection of anti-MAdCAM-1 or negative control monoclonal antibody (mAb). Mice were sacrificed at the indicated age for histological assessment of pancreatic islet inflammation or monitored for diabetes onsets (**b**). (**a**): Distribution and mean ± standard deviation (SD) of islet inflammation score in two treatment groups. (**b**): Diabetes onset in two treatment groups. Log-rank test, ** *p* < 0.01 compared to control negative control mAb treatment.

**Figure 5 ijms-25-11350-f005:**
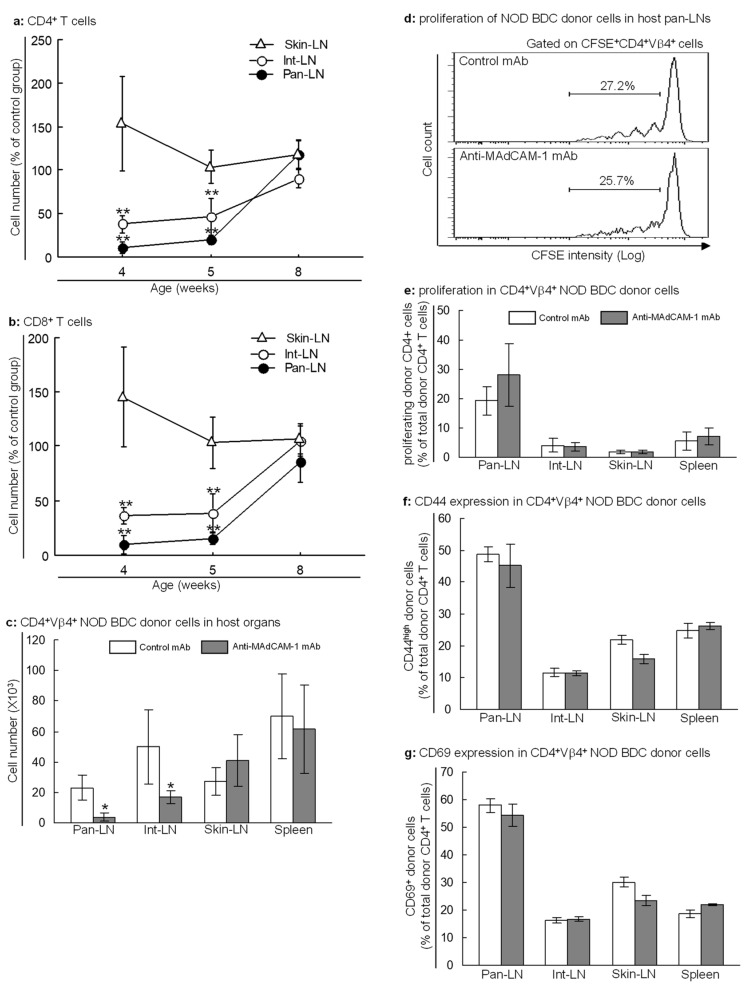
Blockade of MAdCAM-1 in young nonobese diabetic (NOD) mice does not block the priming of autoreactive T cells in pancreas-draining lymph nodes (pan-LNs). (**a**,**b**): Three-week-old female NOD mice were given an intraperitoneal injection of anti-MAdCAM-1 or negative control monoclonal antibody (mAb). Mice were sacrificed at 4, 5 and 8 weeks of age and the absolute numbers of CD4^+^ T cells (**a**) and CD8^+^ T cells (**b**) were determined by immunofluorescence cell suspension staining and flow cytometric analysis. All data are mean ± standard deviation from *n* = 3–5 mice in each group. Two-way analysis of variance (ANOVA) followed by two-sample comparison, ** *p* < 0.01 as compared to isotype-matched negative control mAb-treated mice at same time points. (**c**–**g**): 5(6)-carboxyfluorescein diacetate succinimidyl ester (CFSE)-labeled lymphocytes from 3–4-week-old NOD/BDC 2.5 mice were transferred intravenously into anti-MAdCAM-1- (gray bars) or control mAb- (white bars) treated 3-week-old NOD mice. More than 95% of the donor autoreactive CD4^+^ T cells were naive, as defined by CD44^low^CD45RB^high^. Host mice were sacrificed one week after cell transfer. The absolute number (**c**), proliferation (**d**) identified by 5(6)-carboxyfluorescein diacetate succinimidyl ester (CFSE) intensity reduction, and activation (**e**) identified by CD44^high^ and CD69^+^) of donor-derived autoreactive CD4^+^ T cells in LNs and spleens of anti-MAdCAM-1 or negative control mAb-treated NOD host mice were analyzed by immunofluorescence cell suspension staining and flow cytometric analysis. All data are mean ± standard deviation from *n* = 3–4 mice/group. Two-way ANOVA followed by two-sample comparison (**c**) or two-sample unpaired Student’s *t*-test (**d**–**g**), * *p* < 0.05 compared to the control mAb group.

**Figure 6 ijms-25-11350-f006:**
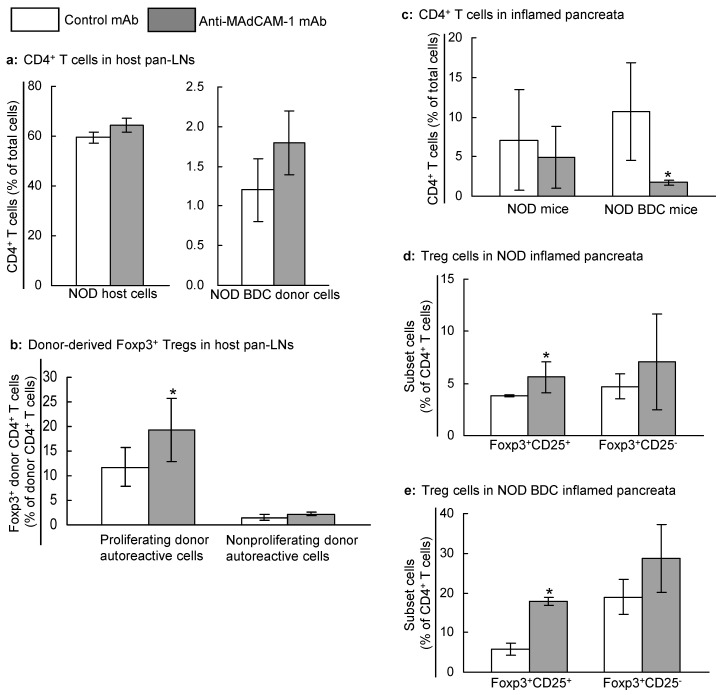
Blockade of MAdCAM-1 in young NOD and NOD/BDC2.5 mice is associated with increased autoantigen-specific regulatory CD4^+^ T cells (Tregs) in pancreas-draining lymph nodes (pan-LNs) and inflamed pancreas. (**a**,**b**): CFSE-labeled lymphocytes from 3–4-week-old NOD/BDC 2.5 mice were transferred intravenously into anti-MAdCAM-1- or control monoclonal antibody (mAb)-treated 3-week-old female NOD mice. One week after cell transfer, host pan-LNs were analyzed for the proportions of donor and host CD4^+^ T cells (**a**) and for Foxp3-expressing Tregs in proliferating (**b**, left) and nonproliferating (**b**, right) donor-derived cells. (**c**–**e**): Lymphocytes were harvested from pancreata of 16-week-old NOD mice (**c**,**d**) and 5-week-old NOD/BDC2.5 mice (**c**,**e**) that received anti-MAdCAM-1 or control mAb at 3 weeks of age. In both strains, MAdCAM-1 blockade reduced the percentage of total CD4^+^ T cells (**c**) and increased the percentages of Treg phenotype (Foxp3^+^CD25^+^ or Foxp3^+^CD25^−^) CD4^+^ T cells in NOD (**d**) and NOD/BDC2.5 (**e**). Data are mean ± standard deviation from 3–5 mice/group. Two-sample unpaired Student’s *t*-test, * *p* < 0.05 compared to negative control mAb. Foxp3: Forkhead box protein P3.

**Table 1 ijms-25-11350-t001:** PNAd staining patterns on HEVs in 3–4-week-old female NOD mice.

Organ	Number of Mice	Strong Diffuse (%)	Mixed (%)	Weak Abluminal (%)
Pan-LN	10	17.4 ± 7.3	26.5 ± 5.2	56.9 ± 11.2
Int-LN	10	81.0 ± 13.0 **	12.4 ± 7.2 **	6.7 ± 6.6 **
PP	9	4.8 ± 5.5 **	9.9 ± 4.0 **	85.2 ± 7.6 **
Skin-LN	10	100.0 ± 0.0 **	0.0 ± 0.0 **	0.0 ± 0.0 **

%: staining pattern distribution in all HEVs from 9–10 mice for each tissue. Unpaired two-sample student’s *t*-test, ** *p* < 0.01 compared to pan-LNs.

## Data Availability

All data are included within this article.

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
