# Peer review of "Mucosal Addressin Cell Adhesion Molecule-1 Mediates T Cell Migration into Pancreas-Draining Lymph Nodes for Initiation of the Autoimmune Response in Type 1 Diabetes"

_ijms, 2024, doi:10.3390/ijms252111350_

Round 1
Reviewer 1 Report
Comments and Suggestions for Authors
This is a very interesting study based on well designed experiments. The manuscript is well written. The introduction has provided sufficient background information for readers to understand this topic. The methods are detailed and easy to follow. The data is presented clearly in the results part. The authors have found that non-obese diabetic mice have high expression of MAdCAM-1 in HEVs in pan-LNs. Blocking MAdCAM-1 can increase Treg in pan-LN and pancreas, and prevent development of diabetes.
Here are my comments:
Please add some details about NOD model in the introduction part so that readers can understand why you choose mice at 3-4 weeks for the experiments.
The authors have shown that blocking MAdCAM-1 during 'critical window' can prevent diabetes in this model. However, in real clinic situation, it is almost unlikely to start such intervention during subclinical phase of the illness. Have the authors had any data if the treatment were started at a later time when the mice have had diabetic sign? If not, please discuss this in the discussion part.
Author Response
Reviewer #1
Comment #1: Please add some details about NOD model in the introduction part so that readers can understand why you choose mice at 3-4 weeks for the experiments.
Response: Thank you for your suggestion. As suggested, we included a brief introduction on NOD mice in Lines 44-48 as “Non-obese diabetic (NOD) mice is a model for human autoimmune type 1 diabetes characterized by the progressive destruction of insulin-secreting β cells by the infiltration of autoreactive T cells in pancreatic islets, with early onset and high rate of diabetes in female mice (https://www.jax.org/strain /001976, October 15, 2024)” and in Lines 50-51 as “----at 3 weeks of age---”.
Comment #2: The authors have shown that blocking MAdCAM-1 during 'critical window' can prevent diabetes in this model. However, in real clinic situation, it is almost unlikely to start such intervention during subclinical phase of the illness. Have the authors had any data if the treatment were started at a later time when the mice have had diabetic sign? If not, please discuss this in the discussion part.
Response: The present study focused on the role of MAdCAM-1 and its ligands in the
homing of T cells to pancreatic lymph nodes and T1D at the autoimmunity initiation stage.
The influence of blocking MAdCAM-1 and its ligands on T1D has been reported by us and
other investigators. This has been stated in Introduction section, lines 50-54 as “We and
others have shown that mucosal addressin cell adhesion molecule-1 (MAdCAM-1) is highly
expressed on vascular endothelia in inflamed islets of NOD mice and is important for
recruiting α4β7 integrin+ lymphocytes into the islets and for development of T1D[4, 8, 22-24].
In contrast, deficiency of L-selectin, which is the major lymphocyte receptor for endothelial
peripheral node addressin (PNAd), does not affect the development of T1D in NOD
mice[25, 26]”.
Reviewer 2 Report
Comments and Suggestions for Authors
This study investigates the importance of MAdCAM-1 in the development of T1DM. Naive autoreactive T cells are primed by islet antigens in pancreas-draining lymph nodes. However, the adhesion molecules that recruit T cells into the pancreatic lymph nodes are not characterized. In this study, the authors reports data indicating that the endothelial venules in pancreatic lymph nodes have a unique adhesion molecule profile that includes strong expression of MAdCAM-1. The use of antibodies against this molecule blocked more than 80% of the migration of the naive autoreactive CD4+ T cells from blood vessels into the pancreatic lymph nodes. Such an inhibition during the initial stage of the autoimmune response appears to provide long lasting (1 year in mouse life) protection from diabetes. Interestingly, the approach does not prevent islet inflammation, suggestion that the process may activate immuno-suppressive processes associated with the inflammation. In support of this hypothesis, the authors provide evidence that blockade of MAdCAM-1 is associated with a relative increase in Tregs in both pancreas and pancreatic lymph nodes.
Comments: The study is properly conducted and written (just a couple of typos, e.g. line 19). The conclusions are supported by the data presented in the manuscript. The title convey the content of the study properly; the M&M section is properly described; the results can be followed properly and the discussion is appropriate.
Q1: why were female mice almost exclusively used for the study? What was the rationale?
Author Response
Reviewer #2
Comment #1: why were female mice almost exclusively used for the study? What was the rationale?
Response: Thank you for your comment. This has been addressed in Lines 83-86 as “T1D onset occurs early with a high incidence (90-100% by 30 weeks of age) in female as compared to male (40-60% by 30-40 weeks of age) NOD mice (https://www.jax.org/strain/001976#, October 15, 2024). In our NOD colony, peri-insulitis develops at 4-5 weeks of age and ~90% of females are diabetic by 30 weeks. Thus, female NOD mice at 3-4 weeks of age were used for all experiments, unless otherwise stated”.
Reviewer 3 Report
Comments and Suggestions for Authors
The authors present a good analysis of an adhesion molecule profile in NOD mouse LNs/PP demonstrating strong and consistent expression of MAdCAM-1 among different LNs. Using anti-MAdCAM-1 mAb, they demonstrated selective inhibition of naive CD4 T cell migration into such LNs (including the use various adoptive cell transfer methods) associated with early treatment (i.e., young mice 3/4-wk vs. older 10-wk mice) in addition to demonstrating a profound and potent efficacy to prevent diabetes onset. Indeed, this study is of high interest to the scientific and therapeutics industry.
There are a few minor points that could improve the manuscript:
1. Line 126: provide a statement describing how the dose of 500 ug of mAb was derived (this should also support the 30 ug/kg dose stated on line 148 as well).
2. In Figure 1, the legend should state whether each row of 3 images is in fact from the same tissue preparation that was stained differently (which is my assumption)
3. Please comment on the "functional significance" of "abluminal" with respect to addressin expression and what that means for cell immigration and emigration of the LN (perhaps include this in the Discussion summary as well).
4. Table 1:
a. define the "%" designation that appears in the column headings; i.e., does it refer to the percentage of all LNs evaluated (my assumption)
b. why are PNAd data only presented in the table and not MAdCAM-1? even if MAdCAM-1 expression is very consistent among all LNs/PP, this comprehensive data should appear in Table 1 alongside the PNAd data which would be helpful in understanding the unique profiles for each tissue.
5. Line 226: clarify the functional significance of "abluminal" expression of PNAd
6. Lines 227-230: While PNAd expression data is presented for 4 tissues, the authors at this point in the narrative make the statement that they will focus "pan-LNs" expression but then proceed to describe expression in all LNs/PP, and then proceed to make a conclusion (Line 233-236) only about the significance of pan-LN. This entire paragraph should be re-written in such a way to describe the unique differences in PNAd AND Madcam in all tissues prior to making an argument to focus on the pan-LN. This problem also appears later on in the manuscript; e.g., lines 259-266.
7. with respective to the use of T cells from the BDC2.5 mice, the authors should give a rationale for why they are using such cells; i.e., it is for obvious technical reasons to track such cells after adoptive transfer, so state the significance of Vb4 expression, why "6-month old" mice vs. very young NOD, etc... - perhaps this explanation should appear in the methods.
8. Lines 272-274: this conclusion statement is false; i.e., MAdCAM-1 mediates migration is not "organ-selective" toward "pan-LN"
9. Lines 306-308: this efficacy results is very strong and exciting. Why only N=10/group for a female NOD mouse efficacy study? typically, N=20 is common. also, has the study been repeated?
9. Line 351: "At 4 weeks of age,.." should be changed to "after 1 week" for these kinds of adoptive transfer experiments
10. Line 395: the "Foxp3+ CD25- Tregs" were not significant so they should be omitted from the statement
11. Discussion first paragraph: a clearer argument should be made of why "pan-LN" are so important with respect to the unique expression patterns of adhesion molecules relative to the other LN/PP that were evaluated.
